# A Retrospective, Single-Centre Study on the Learning Curve for Liver Tumor Open Resection in Patients with Hepatocellular Cancers and Intrahepatic Cholagangiocarcinomas

**DOI:** 10.3390/ijerph19084872

**Published:** 2022-04-17

**Authors:** Bartlomiej Banas, Piotr Kolodziejczyk, Aleksandra Czerw, Tomasz Banas, Artur Kotwas, Piotr Richter

**Affiliations:** 1First Department of Surgery, Jagiellonian University Medical College, 30-688 Krakow, Poland; piotr.1.kolodziejczyk@uj.edu.pl (P.K.); piotr.richter@uj.edu.pl (P.R.); 2Department of Health Economics and Medical Law, Medical University of Warsaw, 02-091 Warsaw, Poland; aleksandra.czerw@wum.edu.pl; 3Department of Economic and System Analyses, National Institute of Public Health NIH—National Research Institute, 00-791 Warsaw, Poland; 4Department of Gynaecology and Oncology, Jagiellonian University Medical College, 31-501 Krakow, Poland; tomasz.1.banas@uj.edu.pl; 5Department of Radiotherapy, Oncology Centre, Maria Sklodowska-Curie Institute, 30-688 Krakow, Poland; 6Sub-Department of Social Medicine and Public Health, Department of Social Medicine, Pomeranian Medical University, 71-210 Szczecin, Poland; artur.kotwas@pum.edu.pl

**Keywords:** hemihepatectomy, hepatocellular carcinoma (HCC), intrahepatic cholangiocarcinoma (ICC), learning curve

## Abstract

Background: Liver resections have become the first-line treatment for primary malignant tumors and, therefore, are considered a core aspect of surgical training. This study aims to evaluate the learning curve for the safety of open hemihepatectomy procedures for patients suffering from hepatocellular carcinoma (HCC) or intrahepatic cholangiocarcinoma (ICC). Methods: This single tertiary center retrospective analysis includes 81 consecutive cases of right or left hemihepatectmy. A cumulative sum (CUSUM) control chart was used to investigate the learning curve. Results: The CUSUM curve for operative time and blood loss level peaked at the 29th and 30th case, respectively. The CUSUM curve for minor adverse effects (mAEs) and severe adverse effects (sAEs) showed a downward slope after the 27th and 36th procedures; the curve, however, remained within the acceptable range throughout the entire study. Conclusion: When performing open hemihepatectomies in patients with HCC and ICC, the stabilization of the operative time and intraoperative blood loss level are gained earlier than sAEs risk reduction.

## 1. Introduction

Hepatocellurar carcinoma (HCC) and intrahepatic cholangiocarcinoma (ICC) are the two most common primary liver malignancies [1]. Factors which have been recognized as increasing the risk of those type of cancers include, i.e., chronic infection with the hepatitis B virus, exposure to aflatoxin B1, obesity, alcoholism, chronic infection with the hepatitis C virus, diabetes and the metabolic syndrome [2,3].

These two types of cancers usually occur in the more developed countries’ populations [4,5]. In Poland, there were 26,280 cases identified in the country’s male population between 1999 and 2016, with the standardized incidence rate oscillating between 3.78 and 5.51, and the mortality rate between 3.42 and 4.43. In the Polish population of women, the standardized incidence rate in the period analyzed fluctuated between 1.3 to 2.61, with the mortality rate at 4.23 up to 5.73 [6].

Primary liver malignancies show aggressive biology and unfavorable prognosis of recovery [7,8,9]. The basic treatment method of these types of cancer is a radical surgical resection [7,8,10]. Since these cancers are usually diagnosed in an advanced stage, extensive surgical procedures (including right-side or left-side hepatectomies) become necessary [9,10]. Nowadays, more and more non-invasive procedures (i.e., laparoscopic or robot-assisted) are being conducted.

There are already many studies published that analyze the learning curve for liver resection procedures implementing minimally invasive surgery [11,12,13,14,15,16,17]. Gulibaud et al. postulated that the learning curve for laparoscopic liver resections is a time-consuming process, and especially major laparoscopic liver resections must be implemented gradually under the supervision of experienced surgeons. Moreover, training outside the operating room may be helpful in gaining surgical experience and reduce adverse effects connected with these procedures [15]. Luft et al. concluded that liver major resections, when regionalized and limited to high-volume centers such as hospitals performing 200 or more of these operations annually, present significantly lower death rates compared with lower-volume centers [16]. For patients diagnosed with other malignancies than a primary liver malignant tumor (especially colorectal carcinoma), but also with confirmed liver metastases, the Italian Consensus recommends minimally invasive simultaneous resections for synchronous liver metastases [17].

The aim of this paper is a retrospective analysis of the learning curve for advanced liver resection techniques in open surgery, including right and left hemihepatectomy, taking account of patient safety parameters.

## 2. Materials and Methods

### 2.1. Patients and Procedures Characteristics

A retrospective analysis of the medical registry comprising patients who underwent laparotomy and liver tumor resection due to primary HCC or ICC was approved by the Institutional Review Board. The study covered the period 1 January 2010 until 31 December 2020, and all patients received surgery performed by the same surgeon—a specialist in general surgery and in training in hepatobiliary surgery. The inclusion criteria were as follows: (1) age 18 years or above, (2) primary HCC or ICC in histopathological reports, and (3) no previous liver surgeries; cases with missing data were excluded. As described in detail in our prior paper, the end points analyzed were following: (1) operating time measured from skin incision to skin closure; (2) intraoperative blood loss level, defined as blood volume removed by suction; (3) post-operative hospital stay length from the first post-operative day to the hospital discharge date [14]. All of the performed procedures were classified as left or right hemihepatectomies, according to the Brisbane 2000 Terminology of Liver Anatomy and Resections [18], and were considered as major liver resections. Similar to our former study, patient safety was evaluated based on the presence of adverse events (AE), an inevitable aspect of the medical services provided. These events were defined as minor AEs (mAEs), matching grade I-IIIa complications per the Clavien–Dindo classification, and severe Aes (sAEs), corresponding to grade IIIb-Ivb complications [19]. There were no grade V complications, e.g., patient’s death, reported in this study. Grade I-IIIa complications included (1) wound infection, (2) prolonged hospital stay (>10 days), and (3) hematoma managed non-surgically. Grade IIIb-Ivb complications were as follows: (1) patient death, (2) admission to the intensive care unit, (3) reoperation due to intraperitoneal bleeding, (4) wound dehiscence requiring resuturing under general anesthesia, (5) hepatobiliary fistula requiring relaparotomy, and (6) post hepatectomy liver failure (PHLF) [14,19]. PHLF was defined according to the International Study Group of Liver Surgeries (ISGLS) consensus [20]. Preoperative Poratal Vain Embolisation (pPVE) was not performed in the analyzed cohort.

Due to extensive liver resection, in all of the cases that analyzed blood and/or frozen plasma, transfusions were not considered as adverse effects.

### 2.2. Basic Characteristics of Hepatobiliary Surgical Training

The Polish surgical training system does not provide a specialization in hepatobiliary surgery. That is why 30 years ago, our tertiary surgery unit established and then developed a proprietary tutoring program in that procedure. In the late 90’s, one of our experienced general surgeons underwent two-year foreign training in hepatobiliary surgery and achieved skills and competences entitling to independently performed, complex surgical procedures on liver and the biliary tract. Upon his return, he began performing these procedures as well as supervising other doctors training in hepatobiliary surgery. Currently, the program is dedicated to specialists in general, colorectal and oncologic surgery, and is aimed at maintaining the potential of at least three well-skilled and experienced hepatobiliary surgeons in the unit. At present, one surgeon is training at a time and starts operating only after finishing the theoretical tutorial and assisting in at least 20 hepatobiliary procedures. The first 10 small liver resections (e.g., segmentectomy or bisegmentectomy according to the Brisbane 2000 terminology [19]) and the first 15 major liver resections (e.g., right/left hemihepatectomy resection or right/left extended hepatectomy according to the Brisbane 2000 terminology [19]) the trainee performs are under the direct supervision of an experienced hepatobiliary surgeon. Subsequently, they operate without direct control, however, an experienced hepatobiliary consultant who is not a member of surgical team is available in the operating theatre throughout the whole surgery. Such a procedure is safe for the patient, and it also allows the surgeon in training to develop their own competences and gain experience. Before every procedure, each patient is advised about the type of surgery they are about to undergo, the possible risks and complications, as well as the names of surgical team members. In this way, patients’ informed consent is secured. If the patient does not agree to be operated on exclusively by the operating team, the surgery is led by a consultant in hepatobiliary surgery, not the trainee. During the procedure, the operating time and intraoperative blood loss is recorded; incidence of complications and the post-operative stay length is also collected together with the standard medical data.

### 2.3. Statistical Analysis

Similar to our former study which investigated the learning curve for hemihepatectomy in patients with primary liver neoplasms, a cumulative sum control chart (CUSUM) analysis was used to examine the learning curve in terms of operative time as well as intraoperative blood loss level and hospital stay length [12,13,14]. Body mass index (BMI) was calculated by dividing the body mass by the square of the body height and is presented as kg/m^2^. Data are presented as the mean ± standard deviation (SD) or as the median ± standard error of the mean (SEM) depending on their distributions, which were checked using the Kologomorov–Smirnov test. To compare the groups of patients with HCC and ICC, the parametric Student’s *t*-test, the nonparametric Mann–Whitney *U* test and the chi-squared test were used as appropriate. For evaluating categorical data, the chi-squared test was employed. A *p* value of 0.05 was considered statistically significant, and all the calculations were performed using STATISTICA data analysis software, version 12.0 (StatSoft, Inc. (www.statsoft.pl) 2019. STATISTICA data analysis software system, version 12.), and MedCalc Statistical Software (MedCalc Software Ltd.; Ostend, Belgium).

## 3. Results

### 3.1. Patients and Procedure

From the total number of 87 identified cases, 5 were excluded due to missing data, and as a result, 82 cases were analyzed. Patients suffering from ICC were significantly younger and presented lower BMI compared to patients diagnosed with HCC. A total of 68 (82.93%) patients had cholecystectomy performed at the time of liver resection, while 14 (17.07%) were after a former cholecystectomy (11 laparoscopically and 3 with laparotomy). This, however, did not significantly influence the median operating time (275.00 min; IQR: 107.00 vs. 262.00 min; IQR: 165.00; *p* = 0.245), intraoperative blood loss (340.00 mL; IQR: 102.00 vs. 372.00 mL; IQR: 55.00; *p* = 0.325), total hospital stay (9.0 days; IQR: 10.0 vs. 13.00 days; IQR: 7.00; *p* = 0.254) or ICU stay (0.0 days; IQR: 1.0 vs. 0.00 days; IQR: 2.00; *p* = 0.128). Open right hemihepatectomy was performed in 59 (71.95%) cases, while open left hemihepatectomy in 23 (28.05%) cases. Detailed characteristics of the patients are presented in Table 1.

### 3.2. Learning Curve Endpoints

In the whole cohort, the median operating time was 255 min (IQR: 110), the median intraoperative blood was 342 mL (IQR: 154), and the median post-operative hospital stay was 10 days (IQR: 9). There were no significant differences between patients with HCC and ICC concerning the presented variables (see Table 1).

Based on operating time, the CUSUM analysis identified procedure no. 29 as the cut-off point of gaining stable and repeatable surgical experience (Figure 1). Procedures 1–29 were classified as “time early” (tE), while cases 30–82 as “time late” (tL). Median operating time (350.00 min (IQR: 65.00) vs. 220 min (IQR: 80.00); *p* < 0.001) and blood loss level (750.00 mL (IQR: 885.00) vs. 310 mL (IQR: 100.00); *p* < 0.001) were significantly higher in the tE group compared to the tL procedures. Additionally, in the tE cohort, the median postoperative hospital stay was significantly longer compared to the tL hospitalizations (15.00 days (IQR: 10) vs. 7 days (IQR: 6); *p* < 0.001). No significant differences in patients’ mean age and mean BMI were observed between the analyzed groups.

When focusing on the intraoperative blood loss level, the cut-off point for the learning curve was procedure no. 30—this allowed us to distinguish procedures 1–30 as tE and 31–82 as tL. Similarly to previous results, we observed significantly longer operative times (335.00 min (IQR: 65) vs. 222.00 min (IQR: 82); *p* < 0.001), higher level of intraoperative blood loss (775 mL (IQR: 885) vs. 300.00 mL (IQR: 130); *p* = 0.001) and prolonged post-operative hospitalization (15 days (IQR: 10) vs. 7 days (IQR: 5.5); *p* < 0.001) in the tE cohort compared to the tL group. Again, no significant differences in patients’ mean age and mean BMI were observed between the above groups.

### 3.3. Patient Safety

In the entire cohort, 40 (48.78%) minor and 28 (34.15%) severe adverse effects occurred (see Table 1). Two cases of post-hepatectomy liver failure required admission to the intensive care unit, therefore, the total number of patients with sAEs is 26, while the total number of sAEs is 28. Additionally, 23 (28.39%) patients required blood and/or frozen plasma transfusions—these were not analyzed as adverse effects.

Furthermore, the CUSUM analysis showed that the incidence of mAEs decreased after the 26th procedure, which was consistent with the operative time and comparable in intraoperative blood loss level (Figure 2). For sAEs, a stable decrease in sAEs was accomplished after the 37th procedure, which was significantly higher than the operative time and intraoperative blood loss peak points. As expected, mAEs and sAEs occurred more frequently during procedures 1–26 (17/26 (73.91%) vs. 23/59 (38.98%); *p* = 0.001 and 16/26 (61.53%) vs. 10/59 (16.95%); *p* < 0.001, respectively) and 1–39 (25/39 (58.97%) vs. 17/43 (39.53%); *p* = 0.002 and 21/37 (56.76%) vs. 5/45 (11.11%); *p* < 0.001, respectively).

## 4. Discussion

A literature review brought us to a conclusion that, although these procedures are essential for residency training in general and oncologic surgery specializations, there has been only a few studies evaluating the learning curve for open liver surgery, and what is more, the studies carried out so far usually compare these with the laparoscopic or robotic approach [14,21,22,23,24,25]. Since hemihepatectomies—robotic, laparoscopic, and open—are univocally considered as major liver surgeries, only these procedures were analyzed in our study [21,22]. Based on operating time and intraoperative blood loss analysis, we demonstrated comparable learning curves for open hemihepatectomy in patients with HCC and ICC. The results we obtained are similar to our previous findings on learning curves in the management of metastatic liver tumors of colorectal cancer with open liver surgery [14]. In that paper, we showed that operating time and intraoperative blood loss stabilized after 26th and 24th major liver resection procedures [14]. These findings indicate that the type of liver malignancy is not the most essential factor determining the learning curve in open liver surgery.

However, in an investigation of 299 patients with HCC who underwent open liver resection, Navarro et al. showed that the learning curve was maximized after 42 cases [24]. This stands in contrast to our findings, in which operating time stabilization was reached after the 29th and intraoperative blood loss after the 30th procedure. In this study, however, both the laparoscopic and open approaches were analyzed and compared with open liver resections; laparoscopic liver resections had higher liver-related injury and complications levels during the learning phase, which can result in a certain bias. Nevertheless, in our cohort, the mAEs risk reduction was obtained after 36 procedures and that number is similar to the results reported by Navarro et al. [24].

Many authors who investigated the safety of (mainly laparoscopic) liver resection procedures consider the need for blood and/or fresh plasma transfusion as a sAE, which is consistent with the approach used in our previous study, concerning the small and major liver resection safety in patients with metastatic liver tumors [14,25,26,27]. In this study, focusing on major liver resections in patients with primary cancers, we did not consider any type of transfusion as a sAE.

The main strength of this study is the large number of consecutive cases of hemihepatectomy procedures performed in patients with primary liver malignancies. Additionally, separate operating time, intraoperative blood loss, and AEs analyses provide a new insight into the process of acquiring surgical skills and experience. We are, however, aware that the present series had several limitations. Firstly, this report was based on the experience of one surgeon with some already established surgical techniques. Some authors state that when studying the learning curves for open surgical techniques, one should evaluate a whole team rather than a single surgeon. Nishikimi et al., for instance, postulated that surgery for advanced ovarian cancer cannot be performed by a single surgeon as it involves a wide operative field in the peritoneal cavity and requires assistant surgeons [28]. There are similar conditions in open liver surgery procedures, especially when major resection is performed. Secondly, a retrospective analysis may be biased due to unknown or unidentified factors that, under some circumstances, can influence the final results.

## 5. Conclusions

Hemihepatectomy procedures present similar learning curves to achieve stabilization in operating time and intraoperative blood loss.Operative time and intraoperative blood loss cannot be, however, considered as surrogates for the risk of grade IIIb-IVb complications, according to the Clavien–Dindo classification, as they present significantly different learning curves [19].Learning curves for major liver resections due to metastatic tumors are similar to our findings; therefore, the type and character of a liver tumor (primary or metastatic) does not seem to influence the learning curve.

## Figures and Tables

**Figure 1 ijerph-19-04872-f001:**
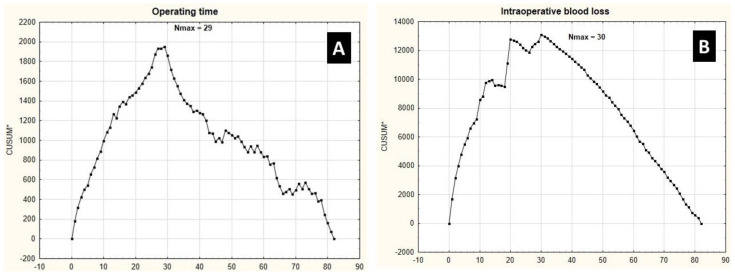
Cumulative sum control chart of operative (CUSUM*) operating time (**A**) and intraoperative blood loss (**B**) against the number of patients.

**Figure 2 ijerph-19-04872-f002:**
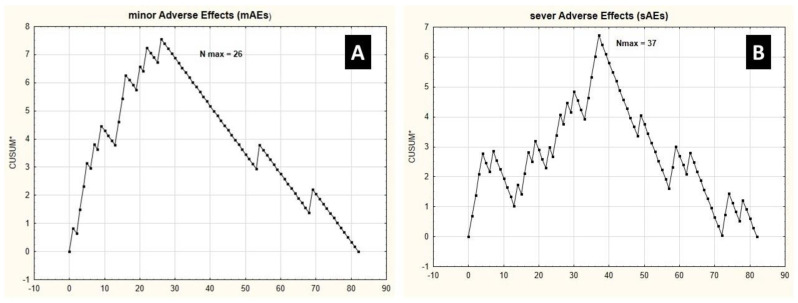
Cumulative sum control chart of operative (CUSUM*) minor adverse effects (mAEs) corresponding with grade I-IIIa complications according to the Clavien–Dindo classification (**A**) and severe adverse effects (mAEs) (**B**) matched with grade IIIb-V complications according to the Clavien–Dindo classification against the number of patients.

**Table 1 ijerph-19-04872-t001:** Baseline patients characteristics to the type of cancer, respectively.

	Total (N = 82)	HCC ^%^ (N = 61)	ICC ^%%^ (N = 21)	*p* *
Mean age (±SD ^$^) [years]	55.07 (±11.54)	60.95 ± 7.32	53.05 ± 12.21	0.006 **
Mean BMI ^$$^ (±SD ^$^) [kg/m^2^]	23.94 (±3.39).	24.52 ± 3.44	22.26 ± 42.64	0.008 **
Male (%)/Female (%)	47 (57.32%)/ 35 (42.68%)	35 (57.38%)/ 26 (45.62%)	13 (59.10%)/ 8(40.90%)	0.716
Median operating time (IQR ^$$$^) [min]	255 (IQR: 110)	250 (IQR: 105)	290 (IQR: 60)	0.245
Median intraoperative blood loss (IQR ^$$$^) [mL]	342 (IQR: 185)	355 (IQR: 310)	340 (IQR: 315)	0.232
Median post-operative hospital stay (IQR ^$$$^) [days]	10 (IQR: 9)	11 (IQR: 9)	6 (IQR: 7)	0.060
Number of grade I-IIIa complications according to the Clavien–Dindo classification (%) wound infection prolonged hospital stay (>10 days) hematoma managed nonsurgically	6 (9.84%) 25 (30.86%) 9 (14.75%)	4 (9.84%) 18 (42.62%) 7 (14.75%)	2 (9.84%) 8 (42.62%) 2 (14.75%)	0.467
Number of grade IIIb-V complications according to the Clavien–Dindo classification (%) Patient death Admission to the intensive care unit Reoperation due to intraperitoneal bleeding Wound dehiscence Hepatobiliary fistula Post-hepatectomy liver failure (PHLF)	0 (0.00%) 14 (17.07%) ^&^ 4 (4.88%) 5 (6.10%) 2 (2.44%) 3 (3.66%) ^&^	0 (0.00%) 10 (16.39%) 3 (4.92%) 3 (4.92%) 2 (3.29%) 2 (3.29%)	0 (0.00%) 4 (19.05%) 1 (4.76%) 2 (9.52%) 0 (0.00%) 1 (4.76%)	0.677
Incidence of Pinard’s maneuver (%)	39 (47.56%)	29 (47.54%)	10 (47.62%)	0.997
Median time of Pringle’s maneuver (IQR ^$$$^) [min]	15 (IQR: 30)	15 (IQR: 30)	15 (IQR: 15)	0.687
Prevalence of preoperative anemia (defined as hemoglobin level < 12.0 and or hematocrit level < 35.0)	28 (34.15%)	20 (32.79%)	8 (38.10%)	0.878
Prevalence of right/left hemihepatectomy ^&&^ (%)	52 (63.41%)/ 30 (36.59%)	41 (67,21%)/ 20 (32,79%)	11 (52.38%) 10 (47.62%)	0.224
Prevalence of liver cirrhosis cirrhosis (%)	59 (62.19%)	50 (81,97%)	9 (42.86%)	0.001 **
Median preoperative ASA ^&&&^ score IQR ^$$$^)	3 (IQR: 1)	3 (IQR:1)	2 (IQR: 1)	0.754

^$^ SD—standard deviation; ^$$^ BMI—body mass index; ^$$$^ IQR—interquartile range; ^%^ HCC—hepatocellular carcinoma; ^%%^ ICC—Intrahepatic cholangiocarcinoma; ^&^ 2 cases of post-hepatectomy liver failure required admission intensive care unit, therefore, the total number of patients with grade IIIb-IVb according to the Clavien–Dindo classification is 26, while the total number of complications grade I-IIIc according to the Clavien–Dindo classification is 28; ^&&^ according to the Brisbane 2000 terminology [19]; ^&&&^ ASA—the American Society of Anesthesiologists Physical Status Classification System; * compared between HCC and IHC; ** statistically significant *p*-value.

## Data Availability

The data presented in this study are available on request from the corresponding author. The data are not publicly available due to restrictions.

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
