# Peer review of "A Retrospective, Single-Centre Study on the Learning Curve for Liver Tumor Open Resection in Patients with Hepatocellular Cancers and Intrahepatic Cholagangiocarcinomas"

_ijerph, 2022, doi:10.3390/ijerph19084872_

Round 1

Reviewer 1 Report

I reviewed the study from Banaset al. The authors presented an interesting analysis of the learning curve for major liver resection with the traditional open approach. As mentioned many papers actually analyze the same topic for minimally invasive surgery. However, the traditional approach needs to be learned by young surgeons. 

I suggest stressing more two aspects in the methods part and in the discussion. 

  • To explain the role of the volume center in the present study. How many resections did the center perform / year and before the study? Did this study is the beginning of the center experience with major resection?
  • Considering the 26/37cases needed to achieve the learning curve, in the last 20 years. nowadays, did the authors think a new young surgeon will need the same number of cases to achieve the curve? ie: in author's center, if you will need to form next surgeon generation in major liver resection, how this experience will impact with timing and case number. 

Author Response

April 11, 2022

Dear Reviewer, 

Please find enclosed a revised version of our manuscript entitled “A retrospective, single-centre study on the learning curve for liver tumor open resection in patients with hepatocellular cancers and intrahepatic cholagangiocarcinomas. (previously submitted under the title: “Learning curve for liver tumor open resection in patients with hepatocellular cancers and intrahepatic cholagangiocarcinomas − use of the cumulative sum method”) (Sub Id ijerph-1650309) which I am re-submitting for consideration for publication in the International Journal of Environmental Research and Public Health

We sincerely appreciated the positive feedback and constructive criticisms of the reviewer and have revised our paper accordingly. Please find a point by point response to the reviewer’s comments below.

Thank you for receiving our paper and considering it for publication. Please feel free to contact me with any questions concerning this manuscript. We look forward to your response. 

Sincerely, 

Bartlomiej Banas, M.D,

Reviewer 1

I reviewed the study from Banaset al. The authors presented an interesting analysis of the learning curve for major liver resection with the traditional open approach. As mentioned many papers actually analyze the same topic for minimally invasive surgery. However, the traditional approach needs to be learned by young surgeons. @ Thank you for the positive feedback.

To explain the role of the volume center in the present study. How many resections did the center perform / year and before the study? Did this study is the beginning of the center experience with major resection?

@ Thank you for this comment. A the beginning of the study we performed 11 small liver resection (sMR) and 8 major liver resection (mLR) per year in patients with primary liver cancers and metastatic live tumours. In the past year the numbers are following 37 for sLR 24 mLR annually. We believe that now we are an experienced high-volume centre for hepatobiliary surgery in the southern Poland.

Considering the 26/37cases needed to achieve the learning curve, in the last 20 years. nowadays, did the authors think a new young surgeon will need the same number of cases to achieve the curve? ie: in author's center, if you will need to form next surgeon generation in major liver resection, how this experience will impact with timing and case number.

@ Thank you very much for this interesting inquiry.  Currently for a trainee in general surgery module, according to the plan of specialisation, it is mandatory to perform 5 procedures involving both liver and spleen including biopsy, resection and trauma management. Additionally 15 procedures on biliary tract including cholecystectomies are required. During training in oncologic surgery performing partial liver resections and liver biopsies are required but the specific numbers of procedures required were not provided. In general surgery specialisation acquisition of following skills: open, laparoscopic liver biopsy; liver abscess drainage, open and laparoscopic segmentectomy; intraoperative ultrasonography is required but again the exact number of above procedures is not specified. These were the reasons for us to perform our study. Nowadays young surgeons are focused on minimal-invasive techniques, mainly laparoscopy but also robotic surgery, that differ from open approach therefore we believe that learning curve for open liver surgery in young surgeons will be comparable to our findings – this however requires confirmation in a novel prospective study.

Reviewer 2 Report

dear Author, thank you for you submission. the topic you treated is actual and interesting, but there are several points i need to fix. 

Introduction should be focused only on liver surgery learning curve (i.e. cite Rocca A, et al.The Italian Consensus on minimally invasive simultaneous resections for synchronous liver metastasis and primary colorectal cancer: A Delphi methodology. Updates Surg. 2021 Aug;73(4):1247-1265. doi: 10.1007/s13304-021-01100-9. Epub 2021 Jun 5. PMID: 34089501.Guilbaud T et al (2019) Learning curve in laparoscopic liver resection, educational value of simulation and training programmes: a systematic review. World J Surg 43(11):2710–2719

Luft HS, Bunker JP, Enthoven AC (1979) Should operations be regionalized? The empirical relation between surgical volume and mortality. N Engl J Med 301(25):1364–1369)

In methods section should be added a dedicated section concerning tutoring and learning curve planning. it is not acceptable to start liver resections without a proper learning program under the supervision of a trained liver surgeon, furthermore it is needed to acquire patient consent and ethical commitee approval for this study, it is not enough the evaluation of the internal reviewer board because the learning program may impact on surgical outcome.

In results paragraph 1 may be resumed in a table.

Liver resections should be classified according to the Brisbane classification.

Indication to open or laparoscopic approach should be better defined

Complications should be classified according to the Clavien Dindo classification

Author Response

April 11, 2022

Dear Reviewer, 

Please find enclosed a revised version of our manuscript entitled “A retrospective, single-centre study on the learning curve for liver tumor open resection in patients with hepatocellular cancers and intrahepatic cholagangiocarcinomas. (previously submitted under the title: “Learning curve for liver tumor open resection in patients with hepatocellular cancers and intrahepatic cholagangiocarcinomas − use of the cumulative sum method”) (Sub Id ijerph-1650309) which I am re-submitting for consideration for publication in the International Journal of Environmental Research and Public Health

We sincerely appreciated the positive feedback and constructive criticisms of the reviewer and have revised our paper accordingly. Please find a point by point response to the reviewer’s comments below.

Thank you for receiving our paper and considering it for publication. Please feel free to contact me with any questions concerning this manuscript. We look forward to your response. 

Sincerely, 

Barlomiej Banas, MD

Reviewer 2

dear Author, thank you for you submission. the topic you treated is actual and interesting, but there are several points i need to fix

@ Thank you for the positive feedback on our study.

Introduction should be focused only on liver surgery learning curve (i.e. cite Rocca A, et al.The Italian Consensus on minimally invasive simultaneous resections for synchronous liver metastasis and primary colorectal cancer: A Delphi methodology. Updates Surg. 2021 Aug;73(4):1247-1265. doi: 10.1007/s13304-021-01100-9. Epub 2021 Jun 5. PMID: 34089501.

Guilbaud T et al (2019) Learning curve in laparoscopic liver resection, educational value of simulation and training programmes: a systematic review. World J Surg 43(11):2710–2719

Luft HS, Bunker JP, Enthoven AC (1979) Should operations be regionalized? The empirical relation between surgical volume and mortality. N Engl J Med 301(25):1364–1369)

@ Thank you for this valuable suggestions  – the Introduction section was supplemented with the above information and relevant references were added.

In methods section should be added a dedicated section concerning tutoring and learning curve planning. it is not acceptable to start liver resections without a proper learning program under the supervision of a trained liver surgeon, furthermore it is needed to acquire patient consent and ethical commitee approval for this study, it is not enough the evaluation of the internal reviewer board because the learning program may impact on surgical outcome..

@ Thank you much for drawing attention to this important issue – the following section was added to the Materials and Methods section:

“2.2 Basic characteristics of hepatobiliary surgical training”

In results paragraph 1 may be resumed in a table.

@ Paragraph 1 was modified according the Reviewer’s valuable suggestions to avoid duplication of the results.

Liver resections should be classified according to the Brisbane classification.

@ Brisbane classification of liver resection was applied as indicated in the review.

Indication to open or laparoscopic approach should be better defined

@ Thank you very much for this important remark.

As indicated in the Introduction most of current research on learning curve in hepatobiliary procedures focus on minimally invasive approach including laparoscopy and robotic surgery. As open surgery is an important issued in surgeons training that cannot be omitted we aimed to investigated the learning curve for advanced liver resection techniques in open surgery, therefore on;y consecutive cases of open approach were eligible for the analysis. The indication for open surgery were following:

  1. Umbilical hernia (n=32)
  2. Liena alba hernia (n=28)
  3. Circulatory insufficiency grade ≥ III according to New York Heart Association (NYHA) classification (n=19)
  4. No patient’s consent for minimally invasive approach (n=3)

Complications should be classified according to the Clavien Dindo classification

@ Thank you very much for this important comment.Surgical complications were reclassified according to Clavien-Dindo classification and presented in our study minor Adverse Effects matched with grade I-IIIb surgical complication according to Clavien-Dindo, while previously described severe Adverse Effects corresponded to grade IIIb-IVb complications. There was no grade V complication reported eg. patient’s death.

Reviewer 3 Report

In this manuscript, the authors aimed at evaluating the learning curve during all consecutive major hepatectomy for hepatocellular carcinoma (HCC) and intrahepatic cholangiocarcinoma (IHCC), performed at a single center by a single surgeon. No statistically significant differences in terms of demographic and  clinical characteristics, as well as of perioperative outcomes, were found between HCC and IHCC patients. Using CUSUM control chart, they found that results in terms of operation duration and intraoperative blood loss started to improve after 29 and 30 liver resections, respectively. In addition, they found that improving results in terms of lower rates of minor and severe adverse events occurred after 26 and 37 liver resections, respectively. Patient operated after the above mentioned cut-offs, compared to those operated before, showed statistically significant better results, in terms of shorter operation duration, lower intraoperative blood loss,  shorter postoperative length of stay, postoperative minor and severe adverse events.

I have some comments:

  • The manuscript contains many misspellings, typos, grammatical and orthographic errors: a review from an English mother tongue Scientific Editor is recommended.
  • I suggest the authors to comply with the guidelines for observational studies reporting (strobe) during the preparation of the manuscript (for example: it should be specified in the title the retrospective single center nature of the study).
  • Introduction:
    • line 50, the authors say ”…of primary malignant liver cancers and cholangiocarcinomas …”, actually cholangiocarcinomas (in particular IHCC) are primary liver tumor and should be considered as primary malignant liver cancers: please correct.
    • Line 55: for stage 4 primary liver tumors (both HCC and IHCC), liver resections is often not indicated.
    • Line 57: “…while resections of only one of the liver lobes are much rarer and non-anatomic resections are almost never made.” The first part of the sentence is not clear: what the authors mean with liver lobe? the second part is uncorrect: non-anatomical resection are usually performed for both HCC and IHCC. Please correct.
  • Material and methods:
    • Starting From Line 74 , the authors repeatedly mention a prior manuscript from their study group: a reference for this manuscript is needed here.
    • Line 98: the authors refer to study groups: which study groups are they speaking about? Please specify.
  • Results: Table 1: I recommend the authors to extend the comparison between HCC and IHCC, including among factors included in the analysis:
    • The side of the liver resected: right hemiliver is volumetrically larger than left hemiliver, which determines increased postoperative risk of developing post-hepatectomy liver insufficiency in case of right hepatectomy, compared to left hepatectomy.
    • The presence of cirrhosis: cirrhosis is much more frequent among patients affected by HCC, compared to IHCC, and its presence increases postoperative risk of developing liver insufficiency.
    • The preoperative asa score: it may surrogate preoperative conditions of the patient.
    • The eventual preoperative use of PVE: it identifies patients with an initial insufficient future liver remnant volume and may act as a surrogate of increased intraoperative difficulty and higher postoperative risk.   
    • Please replace pinard with pringle.
  • Discussion:
    • The authors state that “… we did not consider any type of transfusion as a sAE. That is because, contrary to the patients with metastatic liver tumors, many patients in this study initially suffered from anemia induced either by the cancer or the neoadjuvant treatment.” Actually, concerning HCC preoperative chemotherapy is very rare, while preoperative TACE does not determine anemia. Concerning IHCC too, preoperative chemotherapy is very rarely administered, at least much more rarely than for patients affected by colorectal liver metastases. Then, this sentence makes no sense, please delete or correct accordingly.

Author Response

April 11, 2022

Dear Reviewer, 

Please find enclosed a revised version of our manuscript entitled “A retrospective, single-centre study on the learning curve for liver tumor open resection in patients with hepatocellular cancers and intrahepatic cholagangiocarcinomas. (previously submitted under the title: “Learning curve for liver tumor open resection in patients with hepatocellular cancers and intrahepatic cholagangiocarcinomas − use of the cumulative sum method”) (Sub Id ijerph-1650309) which I am re-submitting for consideration for publication in the International Journal of Environmental Research and Public Health

We sincerely appreciated the positive feedback and constructive criticisms of the reviewer and have revised our paper accordingly. Please find a point by point response to the reviewer’s comments below.

Thank you for receiving our paper and considering it for publication. Please feel free to contact me with any questions concerning this manuscript. We look forward to your response. 

Sincerely, 

Bartlomiej Banas, M.D,

Reviewer 3

In this manuscript, the authors aimed at evaluating the learning curve during all consecutive major hepatectomy for hepatocellular carcinoma (HCC) and intrahepatic cholangiocarcinoma (IHCC), performed at a single center by a single surgeon. No statistically significant differences in terms of demographic and  clinical characteristics, as well as of perioperative outcomes, were found between HCC and IHCC patients. Using CUSUM control chart, they found that results in terms of operation duration and intraoperative blood loss started to improve after 29 and 30 liver resections, respectively. In addition, they found that improving results in terms of lower rates of minor and severe adverse events occurred after 26 and 37 liver resections, respectively. Patient operated after the above mentioned cut-offs, compared to those operated before, showed statistically significant better results, in terms of shorter operation duration, lower intraoperative blood loss,  shorter postoperative length of stay, postoperative minor and severe adverse events.

@ Thank you for your feedback.

The manuscript contains many misspellings, typos, grammatical and orthographic errors: a review from an English mother tongue Scientific Editor is recommended@ Thank you very for this remark – the revised version of the manuscript was underwent an extensive language revision by a scientific editor as recommended

I suggest the authors to comply with the guidelines for observational studies reporting (strobe) during the preparation of the manuscript (for example: it should be specified in the title the retrospective single center nature of the study).

@Thank you for this important comment – the STROBE guidelines were implamentated to the text including title, abstract, methodology, results and discussion and the title was revised appropriately as following: “A retrospective, single-centre study on learning curve for liver tumor open resection in patients with hepatocellular cancers and intrahepatic cholagangiocarcinomas.”

Introduction:

line 50, the authors say ”…of primary malignant liver cancers and cholangiocarcinomas …”, actually cholangiocarcinomas (in particular IHCC) are primary liver tumor and should be considered as primary malignant liver cancers: please correct.

@ Thank you for this remark – the sentence was corrected

Introduction:

Line 55: for stage 4 primary liver tumors (both HCC and IHCC), liver resections is often not indicated.

@ Thank you for this interesting comment – the sentence was rewritten

Introduction:

Line 57: “…while resections of only one of the liver lobes are much rarer and non-anatomic resections are almost never made.” The first part of the sentence is not clear: what the authors mean with liver lobe? the second part is uncorrect: non-anatomical resection are usually performed for both HCC and IHCC. Please correct.

@ Thank you for this valuable remark – the sentence was rewritten

Material and methods:

Starting From Line 74 , the authors repeatedly mention a prior manuscript from their study group: a reference for this manuscript is needed here.

@ Thank you for this remark – the reference was added.

Material and methods:

Line 98: the authors refer to study groups: which study groups are they speaking about? Please specify.

@ Thank you for this comment. The sentence was rewritten in more clear manner as following: “The groups of patients with HCC and ICC were compared using the parametric Student’s t-test, the nonparametric Mann–Whitney U test and the chi-squared test were used as appropriate.” Additionally in line 98 we meant the International Study Group of Liver Surgeries defined in the modified reference 18.

Results:

Table 1: I recommend the authors to extend the comparison between HCC and IHCC, including among factors included in the analysis:

  1. The side of the liver resected: right hemiliver is volumetrically larger than left hemiliver, which determines increased postoperative risk of developing post-hepatectomy liver insufficiency in case of right hepatectomy, compared to left hepatectomy.
  2. The presence of cirrhosis: cirrhosis is much more frequent among patients affected by HCC, compared to IHCC, and its presence increases postoperative risk of developing liver insufficiency.
  3. The preoperative asa score: it may surrogate preoperative conditions of the patient.
  4. The eventual preoperative use of PVE: it identifies patients with an initial insufficient future liver remnant volume and may act as a surrogate of increased intraoperative difficulty and higher postoperative risk.
  5. Please replace pinard with pringle.

@ Thank you for valuable  comment.  Table one was supplemented with information indicated by the Reviewer. Additionally no preoperative Portal Vain Embolization (pPVE) was performed in the analysed cohort – this information was added to the Materials and Methods section.

Discussion:

The authors state that “… we did not consider any type of transfusion as a sAE. That is because, contrary to the patients with metastatic liver tumors, many patients in this study initially suffered from anemia induced either by the cancer or the neoadjuvant treatment.” Actually, concerning HCC preoperative chemotherapy is very rare, while preoperative TACE does not determine anemia. Concerning IHCC too, preoperative chemotherapy is very rarely administered, at least much more rarely than for patients affected by colorectal liver metastases. Then, this sentence makes no sense, please delete or correct accordingly.

@ Thank you for valuable  comment – the sentence was removed.

Round 2

Reviewer 2 Report

Dear Authors, thank you for your changes after reviews. The paper looks fine to me. Best regards.

Reviewer 3 Report

I thank the authors for adequately answering my comments and recommendations.